# Learning Robot Control:
# From Reinforcement Learning to Differentiable Simulation

*Abstract*—**We provide new insights for learning robot control by bridging the gap between learning-centric policy training and model-based control. We leverage principles from optimal control, reinforcement learning, and differentiable simulation to develop control algorithms that enhance the robot's agility while maintaining robustness in real-world scenarios. First, we show that the fundamental advantage of reinforcement learning (RL) in robotics lies in its optimization objective compared to optimal control. Specifically, RL directly maximizes a task-level objective, which can be non-differentiable, whereas optimal control is restricted by the requirement for smooth and differentiable cost functions. The flexibility in objective design allows for achieving more flexible control policies, leading to more robust performance in unexpected scenarios. Second, we propose using policy search to automatically optimize high-level policies for model predictive control (MPC). This formulation enables policy search to focus on maximizing a high-level task objective, while the MPC optimization can concentrate on low-level tracking performance. Third, we explore the potential of differentiable simulation for policy training. Differentiable simulation can provide low-variance first-order gradients, resulting in more stable training and better convergence. We show near-optimal control performance for a toy double integrator and its potential for quadruped locomotion.**

## I. INTRODUCTION

Control systems are at the core of every real-world robot. They are deployed in an ever-increasing number of applications, ranging from autonomous racing and search-and-rescue missions to industrial inspections and space exploration. To achieve peak performance, certain tasks require pushing the robot to its maximum agility. How can we design control algorithms that enhance the agility of autonomous robots and maintain robustness against unforeseen disturbances? This research addresses this question by leveraging fundamental principles in optimal control, reinforcement learning, and differentiable simulation.

Optimal Control [2, 3], such as Model Predictive Control (MPC), relies on using an accurate mathematical model within an optimization framework and solving complex optimization problems online. Reinforcement Learning (RL) [17] optimizes a control policy to maximize a reward signal through trial and error. Differentiable Simulation [16, 6] promises better convergence and sample efficiency than RL by replacing zeroth-order gradient estimates of a stochastic objective with an estimate based on first-order gradients. An overview of these three approaches is summarized in Table I.

Particularly, model-free RL has recently achieved impressive results, demonstrating exceptional performance in various domains, such as quadrupedal locomotion over challenging terrain [11, 12, 14]. Some of the most impressive achievements of RL are beyond the reach of existing optimal control (OC) systems. However, most studies focus on system design; less attention has been paid to the systematic study of fundamental factors that have led to the success of RL or have limited OC.

It is important to highlight that the progress in applying RL to robot control is primarily driven by the enhanced computational capabilities provided by GPUs rather than breakthroughs in the algorithms. Consequently, researchers may resort to alternative strategies such as imitation learning [4] to circumvent these limitations in scenarios where data collection cannot be accelerated through computational means [1, 9, 5, 19]. This highlights the need to study the connection between RL, optimal control, and robot dynamics. We attempt to answer the following three research questions:

**Research Question 1**: *What are the intrinsic benefits of reinforcement learning compared to optimal control?*

**Research Question 2**: *How to combine the advantage of reinforcement learning and optimal control?*

**Research Question 3**: *How to effectively leverage the dynamics of robots to improve policy training?*

## II. REINFORCEMENT LEARNING VERSUS OPTIMAL CONTROL

First, we investigate **Research Question 1** by studying RL and OC from the perspective of the *optimization method* and *optimization objective*. We perform the investigation in a challenging real-world problem that involves a high-performance robotic system: autonomous drone racing. On one hand, RL and OC are two different optimization methods and we can ask which method can achieve a more robust solution given the same cost function. On the other hand, given that RL and OC address a given robot control problem by optimizing different objectives, we can ask which optimization objective can lead to more robust task performance.

Our results indicate that RL does not outperform OC because RL optimizes its objective better. Rather, RL outperforms OC because it optimizes a better objective. Specifically, RL directly maximizes a task-level objective, which leads to more robust control performance in the presence of unmodeled dynamics and disturbance. In contrast, OC is limited by the requirement of optimizing a smooth and differentiable loss function, which in turn requires decomposing the task into planning and control, thus limiting the range of control policies that can be expressed by the system. In addition, RL can leverage domain randomization to achieve extra robustness and avoid overfitting, where the agent is trained on a variety of simulated environments with varying settings.

| Continuous Time Optimal Control Problem | | | |
|---|---|---|---|
| Minimize a cost function over a time horizon: $\min_{x(\cdot),u(\cdot)} \int_0^T \ell(x(t),u(t),t)\,dt + \ell(x(T))$ | | | |
| **Control Method** | **Model Predictive Control** | **Policy Search** | **Backpropagation Through Time** |
| **Optimization Objective** | $J(x,u) = \sum_{k=0}^{N-1} \ell(x_k,u_k) + \ell(x_N)$ | $J(\theta) = \mathbb{E}_{\tau \sim \pi_\theta}\left[\sum_{k=0}^{N} r_k\right]$ | $J(\theta) = \mathbb{E}_{x_0 \sim p(x_0)}\left[\sum_{k=0}^{N-1} \ell(x_k,u_k) + \ell(x_N)\right]$ |
| **Constraints** | s.t. $\begin{cases} x_0 = x_{\text{init}} \\ x_{k+1} = f(x_k,u_k) \\ g(x,u) = 0 \\ h(x,u) \leq 0 \end{cases}$ | - | - |
| **Decision Variables** | $u_k, x_k$ | $\theta$ | $\theta$ |
| **Optimization Method** | Nonlinear Programming | Policy Gradient | Analytical Gradient |
| **Gradients** | $\nabla \mathcal{L}(x,u,\lambda,\mu)$ | $\mathbb{E}_{\tau \sim \pi_\theta}[R(\tau)\nabla_\theta \log p(\tau)]$ | $\nabla_\theta R(\tau)$ |
| **Control Law** | $u_0^*$ | $u \sim \pi_{\theta*}(u|x)$ | $u = \pi_{\theta*}(x)$ |

TABLE I: **Comparison of three methods for approximately solving the continuous-time optimal control problem.**

TABLE. II provides evidence regarding our findings. It highlights the differences in the optimized value functions for two different objectives: Trajectory Tracking and Gate Progress. Trajectory Tracking assigns high values when the state is close to the reference state (x) and low values when it is far away, aligning with its objective of minimizing a quadratic loss function between the vehicle state and its reference. Time allocation of the reference state is the incentive for navigating the drone forward, which is done exclusively during the planning stage. However, pre-computed time-optimal trajectories cannot account for model mismatches in real systems.

Instead, a gate progress maximization reward is not limited by a reference trajectory, allowing for a control behavior that emerges directly from learning to optimize the high-level goal. The learned value function in Gate Progress assigns high values to safe and valid states, such as those near the optimal path, and low values to risky states, such as those near the gate border. Unlike Trajectory Tracking, where deviations from the reference state are penalized, Gate Progress allows the vehicle to adapt its behavior freely during deployment. This adaptability leads to more robust performance when facing unexpected disturbances and model mismatches.

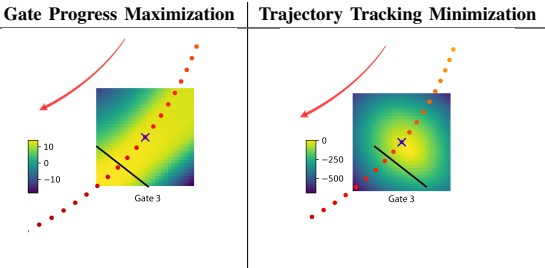

TABLE II: **Comparison of the state value functions for different optimization objectives in drone racing.** The value function is obtained by optimizing two objectives: Trajectory Tracking and Gate Progress. The Gate Progress objective leads to task-aware behavior, such as avoiding gate collisions, which is not observed in Trajectory Tracking.

Our findings allow us to push an extremely agile drone to its maximum performance using RL, achieving a peak acceleration greater than 12g and a peak velocity of $108\ \mathrm{km\,h^{-1}}$. We show that the RL-based neural network policy outperforms state-of-the-art OC-based methods in terms of robustness and lap time because RL does not rely on pre-computed trajectory or path. Fig 1 displays time-lapse illustrations of the racing drone controlled by our RL policy in an indoor flying arena.

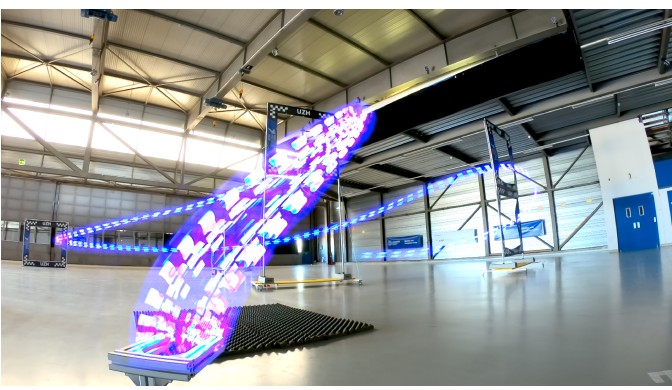

Fig. 1: RL pushes a super agile drone to its physical limit.

## III. POLICY SEARCH FOR MODEL PREDICTIVE CONTROL

Second, we investigate **Research Question 2** by presenting a *policy-search-for-model-predictive-control* framework for merging learning and control. Policy Search and Model Predictive Control (MPC) are two different paradigms for robot control: policy search has the strength of automatically learning complex policies and directly optimizing task objectives using data, while MPC can offer precise control performance using models and trajectory optimization.

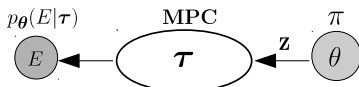

Fig. 2: Graphical model of policy search for MPC.

A visualization of the framework is given in Fig 2. We consider model predictive control (MPC) as a parameterized controller and formulate the search for hard-to-optimize decision variables as a probabilistic policy search problem. Given the predicted decision variables, MPC solves an optimization problem and generates control commands for the robot. A key advantage of our approach over the standard MPC formulation is that the high-level decision variables, which are difficult

to optimize simultaneously with other state variables, can be learned offline and selected adaptively at runtime.

We treat MPC as a controller $\boldsymbol{\tau} = \text{MPC}(\mathbf{z})$ that is parameterized by the high-level decision variables $\mathbf{z}$. Here, $\boldsymbol{\tau} = [\mathbf{u}_h, \mathbf{x}_h]_{h \in 1, \cdots, H}$ is a trajectory generated by MPC given $\mathbf{z}$, where $\mathbf{u}_h$ are control commands and $\mathbf{x}_h$ are corresponding states of the robot. By perturbing $\mathbf{z}$, MPC can result in completely different trajectories $\boldsymbol{\tau}$. To find the optimal trajectory for a given task, the optimal $\mathbf{z}$ has to be defined in advance. First, we model $\mathbf{z}$ as a high-level policy represented by a probability distribution, specifically a parameterized Gaussian distribution. Then, we optimize the policy using probabilistic policy search (or probabilistic inference) algorithms. This leads to the maximum likelihood problem [7]:

$$\max_{\boldsymbol{\theta}} \quad \log p_{\boldsymbol{\theta}}(E = 1) = \log \int_{\boldsymbol{\tau}} p(E|\boldsymbol{\tau}) p_{\boldsymbol{\theta}}(\boldsymbol{\tau}) d\boldsymbol{\tau},$$

which can be solved efficiently using Monte-Carlo Expectation Maximization.

We validate this framework by focusing on a challenging problem in agile drone flight: flying a quadrotor through fast-moving gates. Flying through fast-moving gates is a proxy task to develop autonomous systems that can navigate the vehicle through rapidly changing environments. Our controller achieved robust and real-time control performance in both simulation and the real world.

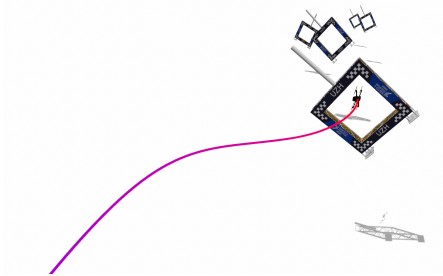

Fig. 3: Flying through a dynamic gate in simulation.

## IV. POLICY LEARNING VIA DIFFERENTIABLE SIMULATION

Third, we investigate **Research Question 3** by demonstrating the effectiveness of differentiable simulation for policy training. Differentiable simulation promises faster convergence and more stable training by computing low-variance first-order gradients using the robot model, but so far, its use for robot control has remained limited to simulation [18, 15, 10, 8]. In differentiable simulation for policy learning, the backward pass is crucial for computing the analytic gradient of the objective function with respect to the policy parameters. Following [13], the policy gradient can be expressed as follows

$$\nabla_{\theta} \mathcal{L}_{\theta} = \frac{1}{N} \sum_{k=0}^{N-1} \left( \sum_{i=1}^{k} \frac{\partial l_k}{\partial x_k} \prod_{j=i}^{k} \underbrace{\left( \frac{\partial x_j}{\partial x_{j-1}} \right)}_{\text{dynamics}} \frac{\partial x_i}{\partial \theta} + \frac{\partial l_k}{\partial u_k} \frac{\partial u_k}{\partial \theta} \right),$$

$$(1)$$

where the matrix of partial derivatives $\partial x_j / \partial x_{j-1}$ is the Jacobian of the dynamical system $f$. Therefore, we can compute the policy gradient directly by backpropagating through the differentiable physics model and a loss function $l_k$ that is differentiable with respect to the system state and control inputs. A graphical model for gradient backpropagation in policy learning using differentiable simulation is given in Figure 4.

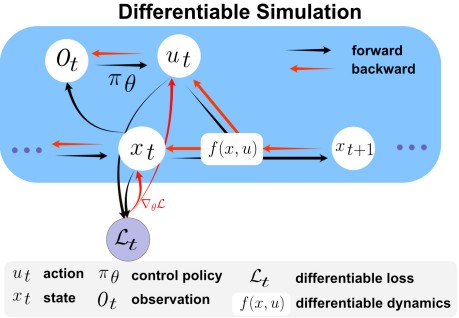

Fig. 4: Graphical model of Differentiable Simulation.

We begin by examining a toy example: control of a double integrator, which is a fundamental problem in system and control theory. The double integrator is a second-order control system that models simple point mass dynamics in one-dimensional space. The state variables are position $x$ and velocity $\dot{x}$, with control inputs $u = \ddot{x}$. The objective is to stabilize the system at the origin, $[x, \dot{x}] = [0, 0]$. Our study compares training a 2-layer multilayer perceptron via differentiable simulation and PPO against a Linear Quadratic Regulator (LQR) controller. We show that DiffSim attains nearly optimal performance with limited training iterations and samples. In contrast, even when scaling the number of simulation environments to 1024 and training the policy with considerably more iterations, PPO fails to achieve the same level of control performance as LQR or DiffSim. This finding suggests that scaling might be insufficient for achieving optimal control using PPO.

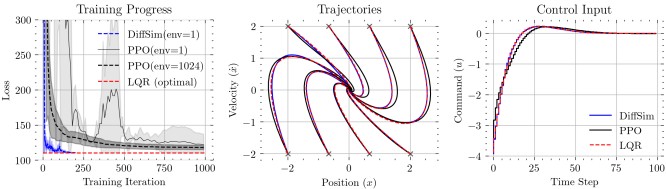

Fig. 5: **Control of a double integrator using LQR, reinforcement learning, and differentiable simulation.** (left): Learning curves. (middle): Trajectories of different control policies. We initialize the system at the same states for all methods. (right): Control inputs generated by different control methods for one specific starting state. The neural control policy trained using differentiable simulation (DiffSim) achieves near-optimal control performance.

Differentiable simulation is an effective approach when the system has continuous and smooth dynamics. The main challenge with differentiable simulation lies in the complex optimization landscape of robotic tasks due to discontinuities in contact-rich environments, such as quadruped locomotion.

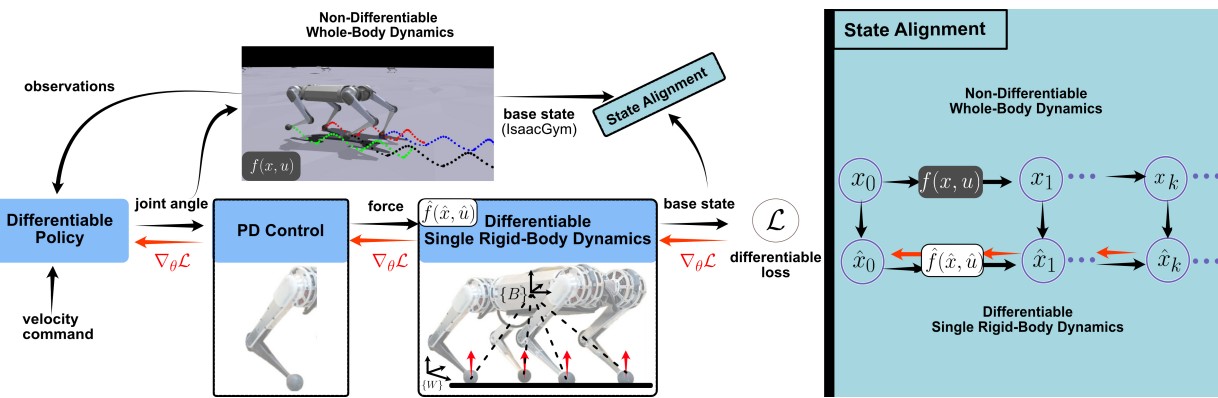

Fig. 6: **System overview of learning quadruped locomotion using differentiable simulation.** Our approach decouples the robot dynamics into two separate spaces: joint and floating base spaces. We leverage the differentiability and smoothness of a single rigid-body dynamics for the robot's main body, which takes the ground reaction force from its legs as the control inputs. Additionally, we treat PD control as a differentiable layer in our computation graph. Finally, we use the state from a more accurate, non-differentiable simulator (IsaacGym) to align the base state in the single rigid-body simulation.

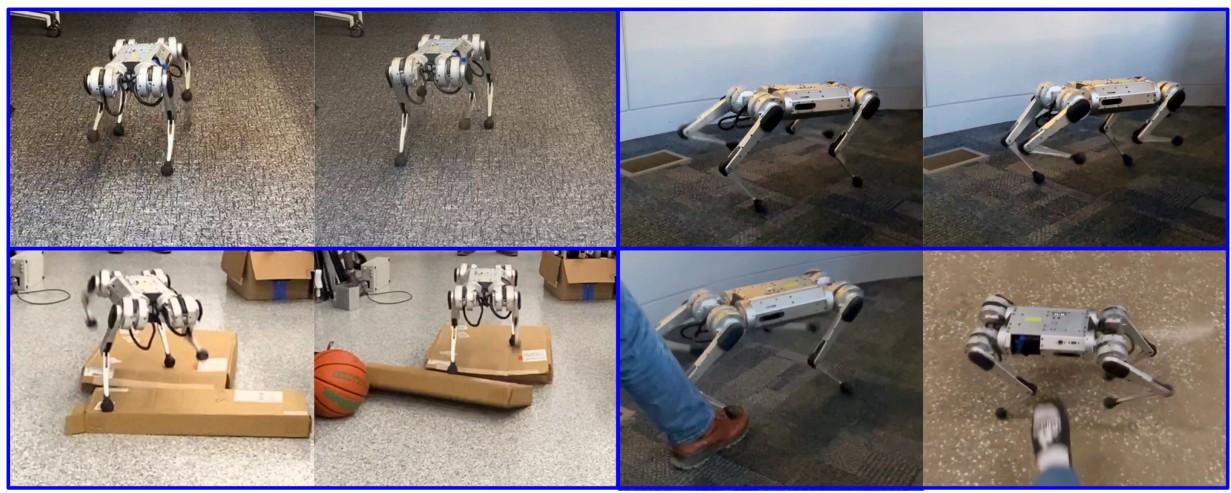

Fig. 7: **Real-world deployment using a Mini Cheetah.** All the experiments are conducted using one single blind policy.

We propose a new, differentiable simulation framework to overcome these challenges. The key idea involves decoupling the complex whole-body simulation, which may exhibit discontinuities due to contact, into two separate continuous domains. Our framework enables learning quadruped walking in minutes using a single simulated robot without any parallelization. When augmented with GPU parallelization, our approach allows the quadruped robot to master diverse locomotion skills, including trot, pace, bound, and gallop, on challenging terrains in minutes. Additionally, our policy achieves robust locomotion performance in the real world zero-shot. To the best of our knowledge, this work represents the first demonstration of using differentiable simulation for controlling a real quadruped robot.

We demonstrate the performance of our policy in the real world using Mini Cheetah. Fig. 7 shows several snapshots of the Robot's behavior using different gait patterns or over different terrains. We trained a blind policy in simulation using 64 robots and then transferred the policy directly to the

real world without fine-tuning. The Robot can walk forward and backward with different gait patterns and frequencies. Moreover, the policy proved robust, enabling the robot to manage certain disruptions, such as unexpected forces applied to its body and locomotion on deformable objects.

## V. CONCLUSION

This work has presented three independent projects focused on learning robot control, providing valuable insights into robotics. Future work could incorporate structured knowledge from robot dynamics and constraints from optimal control into reinforcement learning, potentially reducing the sample complexity and improving learning efficiency. Additionally, policy learning with differentiable simulation could benefit from Temporal Difference (TD) learning, as it enables the combination of first-order optimization methods with model-free TD learning.

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
