# OpenReview forum: "Learning Robot Control: From Reinforcement Learning to Differentiable Simulation"
_roboticsfoundation.org/RSS/2024/Workshop/EARL — EARL 2024 Poster_

### Official Review · Reviewer_qsmN · 2024-06-23

**Rating:** 7
**Confidence:** 3

**Review:**

**Summary:**

The paper presents three optimal control methods: Model Predictive Control (MPC), Reinforcement Learning (RL), and Differentiable Simulation. It discusses their fundamental differences and how each method can be leveraged to optimize control tasks.

Firstly, the paper demonstrates that RL outperforms MPC in optimal control tasks, as evidenced by experiments and visualizations of value functions. This superiority is attributed to RL's task-oriented objective functions. An example task is drone racing through fixed gates.

Secondly, the authors introduce a policy-search-for-model-predictive-control approach, which combines the precise control of MPC with policy search to create a parameterized MPC. This method shows promising results in controlling a drone over fast-moving gates.

The final part introduces differentiable simulation for optimal control. The authors show that this method is generally more sample efficient than RL in tasks like the double integrator. However, they highlight that differentiable simulations struggle in contact-rich environments that produce non-differentiable gradients. To address this limitation, they propose splitting a quadruped robot into two differentiable parts: a rigid body and an approximate leg model represented by forces. This approach achieves good training results and is applied to a real robot.

**Discussion:**

The paper aims to provide insights into state-of-the-art methods for optimal control, offering valuable contributions to the community. It includes a significant number of experiments to support its claims and discusses both the fundamental differences and limitations of these methods, proposing solutions through case examples.

Additionally, good results are shown through their policy-search-for-model-predictive-control for both simulation and real world for drone racing over moving gates. Additionally, the model splitting of the quadruped robot is a good contribution and shows the first usage of differentiable simulation over a real-world robot in this specific task.

**Limitations:**

- In the second and third parts, while the authors claim good results for their models, more qualitative results would be appreciated. However, this lack of qualitative results does not significantly impact the validity of the paper's statements.
- Although each method is valuable, describing them using entirely different tasks (such as drones and quadrupeds) makes the connections between the parts less straightforward, potentially reducing the paper's readability.

**Minor Writing Issues:**

- Figure 6 is not cited in the text.

---

### Decision · Program_Chairs · 2024-06-24

Accept (Poster)